# To Be a Pregnant Surgeon—Is There Anything to Be Afraid of?

**DOI:** 10.3390/ijerph20032265

**Published:** 2023-01-27

**Authors:** Natalia Dowgiałło-Gornowicz, Jakub Jan Zięty, Michał Gornowicz, Klaudia Sztaba, Karolina Osowiecka, Paweł Lech

**Affiliations:** 1Department of General, Minimally Invasive and Elderly Surgery, Collegium Medicum, University of Warmia and Mazury, Niepodległosci 44 St., 10-045 Olsztyn, Poland; 2Department of Economic Law and Commercial Law, Faculty of Law and Administration, University of Warmia and Mazury in Olsztyn, Obitza 1 St., 10-725 Olsztyn, Poland; 3Department of Psychology and Sociology of Health and Public Health, School of Public Health, University of Warmia and Mazury in Olsztyn, Warszawska 30 St., 11-041 Olsztyn, Poland

**Keywords:** pregnant surgeon, pregnancy, female surgeon, maternity law

## Abstract

Background: Women who decide to become a surgeon are afraid of motherhood. The aim of this study was to establish the opinions of patients and doctors on the professional activity of pregnant surgeons (PS). Methods: The study was conducted on a group of respondents consisting of doctors and patients. The study was carried out using a questionnaire of 12 questions. Results: 1074 doctors and 657 patients responded to the survey. Doctors, especially non-surgeons, significantly more often believed that PS should stop working in the operating theatre immediately after pregnancy confirmation. Most patients thought that operations performed by PS are normal, whereas the doctors more often considered it heroic or irresponsible. Doctors more often mentioned fear of financial stability and fear of losing their reputation as reasons for working by PS. Most respondents claimed that it made no difference whether they were operated on by PS or not. However, patients significantly more often declared their willingness to be operated on by PS. Conclusions: The study showed that female surgeons may have slight concerns about how they will be perceived by colleagues and patients. However, most respondents, patients significantly more often, believed that working during pregnancy is the natural course of things.

## 1. Introduction

General surgery is associated with male dominance. In the past, it was identified with strength and long hours of physical work. The beginnings of minimally invasive surgery changed its perception. This may result in the growing trend in the number of women choosing this specialty [1,2]. New problems have arisen as a result.

Women who decide to become a surgeon are afraid of motherhood [3,4]. The main problem is getting pregnant and being pregnant. Women can be afraid to tell their superiors about the pregnancy [3]. They want to avoid stigmatization and removal from work [5,6]. It has been proven that some instruments and operating environment have a negative impact on pregnancy, so they may pose a risk to unborn children [7,8]. We should take into account not only physical but also biological and chemical factors [9,10]. The first issue is the long-term work and stress associated with the profession of a surgeon. Knieper et al. shows that 2% of women experienced a work-related preterm birth, although they are statistically consistent with the same risk in the general population [11]. The toxic effects of exposure to radiation on the developing fetus have been confirmed, and the employer should provide appropriate working conditions for a pregnant woman [7,8]. However, recent studies show that even working in harmful conditions, such as radiation exposure, can be safe for pregnant women, provided that the work is performed under certain conditions and limited as much as possible to be safe [12,13]. The development of minimally invasive techniques has made it possible to perform operations in safe conditions, using devices for coagulation without radiation—for example, using ultrasonic energy [14]. Unlimited quantities of waste anesthetic gases may cause spontaneous abortion or congenital abnormalities [7]. Working in a modern operating room ventilation system reduces operating fumes and anesthetic gazes [7,15]. Therefore, surgery during pregnancy may be safe if the right conditions are met. Another problem is the assessment of a pregnant woman by society, both by colleagues and patients. Women may fear losing their patients. The statistics show that these problems cause surgeons to decide on motherhood several years later than other women [16,17].

The aim of this study was to establish the opinions of patients and doctors on the professional activity of pregnant surgeons (PS) and to estimate the correlations between their opinions and demographic factors.

## 2. Materials and Methods

The study was conducted on a group of respondents consisting of doctors and patients. The study was carried out in Olsztyn, Poland between 1 March and 30 April 2022. The study was conducted using a questionnaire prepared specifically for this study under general principles [18]. The study was approved by the Hospital’s Research Committee. The study was anonymous and voluntary. Completing the questionnaire was tantamount to signing the consent to participate in the study. The questionnaire was developed in Google Forms. It was an open survey for each visitor of a site. The respondents were doctors working in Polish health services and patients treated in Poland. The survey was distributed via social media—Facebook and Instagram. On Facebook, it was advertised to a Polish group of young residents and specialists, female doctors (to be a part of the group, providing the license number of the right to practice is required), and patients of the Municipal Polyclinical Hospital in Olsztyn. On Instagram, it was advertised by a Polish doctor (see Acknowledgments). The questionnaire consists of 12 questions, including 6 main closed-ended questions and 6 demographic questions (age, sex, place of residence, education/specialization, having children, being in relationship), Appendix A. Each section was on a separate Google Forms page.

The differences between doctors and patients due to demographics and opinions related to actively working pregnant surgeon was assessed using the chi-square test (for categorical variables) and Mann–Whitney test (for continuous variables). The normal distribution of variables was tested by Shapiro–Wilk test. A *p*-value of < 0.05 was considered to be significant. The analysis was conducted using Statistica software, version 13. htp://statistica.io accessed on 20.05.2022 TIBCO Software Inc., Krakow, Poland (2017).

## 3. Results

A total of 1731 people responded to the survey. There were 1074 doctors (62%) and 657 patients (38%) (Table 1). Most respondents were women (97.9%, 93%, respectively). Of the patients, 82.6% had higher education, and 39.9% lived in a city with more than 150,000 inhabitants. A total of 66.5% of doctors lived in a large city, 62.9% had a non-surgical specialization, and 87% worked in public health services. A total of 87.2% of doctors and 67.9% of patients had at least one child. A total of 95.1% doctors and 89.6% of patients were in the relationship. A total of 17.7% of doctors and 15.6% of patients stopped working immediately after pregnancy confirmation. A total of 14.7% of surgeons operated until the end of pregnancy, Table 1.

### 3.1. Opinions of Doctors and Patients

Doctors believed—significantly more often than patients—that PS should stop working in hospital and in the operating theatre immediately after confirming a pregnancy (*p* < 001). A total of 77.1% of patients thought that operations performed by PS are normal, whereas the doctors more often considered it heroic or irresponsible (*p* < 0.001). Doctors, compared to patients, more often mentioned fear of financial stability (77.3%) and fear of losing their reputation (22.7%) as reasons for PS continuing to work (*p* < 0.001). Most respondents claimed that it made no difference whether they were operated on by PS or not. However, patients significantly more often declared their willingness to be operated on by PS (93.0% vs. 89.5%, respectively; *p* = 0.01), Table 2.

### 3.2. Opinions of Surgeons and Non-Surgeons

The analysis included 676 non-surgeons (63%) and 398 (37%) surgeons. Non-surgeons significantly more often than surgeons believed that PS should stop working in operating theatre immediately after a confirmation of pregnancy (38.6% vs. 29.4%, respectively, *p* = 0.002). There were no significant differences in working in hospital or outpatient clinic, and most respondents thought it depended. Surgeons more often claimed that operating during pregnancy is normal (55.8% vs. 47.3%, respectively), whereas for non-surgeons, it was more frequently considered heroic or irresponsible (*p* = 0.03). Most surgeons and non-surgeons reported that PS worked during pregnancy for fear of losing financial stability. Significantly more often surgeons reported that PS decide to work for fear of losing their patients (*p* < 0.001). In general, both non-surgeons and surgeons would have no doubts about being operated on by PS (Table 3).

### 3.3. Correlation between Demographic Factors and Respondents’ Answers

Among the patients, men were more likely to believe that PS should stop working in hospitals and outpatient clinics immediately after confirming a pregnancy (p = 0.03). Patients with more than three children significantly more often indicated that PS should resign from work in hospital and outpatient clinic compared to patients with fewer or no children. 

In the group of doctors, men believed that PS should stop working in outpatient clinics immediately after confirming a pregnancy (p = 0.04). Doctors without children significantly more often reported that PS should stop working in hospital after confirming a pregnancy (p = 0.04). Female doctors significantly more often than men indicated that they had no doubts that they could be operated on by PS (89.8% vs. 73.9%, respectively). 

There were no significant correlations between the opinion of patients and doctors and other demographic factors (education, place of residence, being in relationship), Table 4.

## 4. Discussion

Our study is an analysis of the survey about PS conducted on 1074 doctors and 657 patients. In the respondents’ opinion, pregnancy among surgeons does not have to be a reason for resignation from work in an operating theatre. In some ways, the opinion of doctors was more critical than that of the patients. 

Pregnancy delays of up to 7 or 9 years are common among surgical specialties [16,17]. Fear of losing financial stability was the most common answer in our survey among surgeons. Protection of women’s work related to maternity is an element emphasized not only by European Union or Member States’ law, but also in guidelines issued based on the provisions of directives and the case law of The Court of Justice [12,19,20]. Maternity and the related special protection of a woman is comprehensively defined in the EU legal order in the sense that it covers protection during pregnancy, maternity leave, and breastfeeding [19]. The protection of a pregnant woman is not only her right but also the employer’s duty. [21].

It is prohibited to force a pregnant woman to work in conditions of revealed risk that may affect pregnancy or to work at night [21]. The employer is prohibited from performing certain works and cannot assign them to a pregnant woman if she is aware of her condition. On the other hand, in the case of works not covered by such a ban, the possibility of their implementation is associated with an assessment of their impact on the condition of the pregnant woman or fetus. In this case, it is not only important to assess the risk, but also to define measures to counteract its occurrence. As for the analyses conducted by the employer, there may be different assessments of their results by the employee and the employer. The differences may concern not only the issue of the employer not considering all risks, but also of indicating too many of them, or the proposed method of reducing them. A pregnant woman, apart from in cases of statutory prohibition, does not have to be restricted in the performance of her work. It is guaranteed not only by directive, but also by the prohibition of direct discrimination on grounds of sex [19,21]. Even the state of pregnancy cannot be discriminated against in employment. To avoid such situations, the assessment must also consider the possibility of performing work with the introduction of specific preventive measures. The obligation to apply these measures rests with the employer when a pregnant employee expresses the will to perform activities falling within the scope of her duties and there are no medical obstacles to continuing her work.

European legislation allows time off for examinations during pregnancy without any loss of pay [19,20]. On the other hand, in the case of being pregnant, if a woman—a full-time employee—cannot work due to her health condition, the labor law ensures that financial continuity is maintained [19,21]. However, the situation changes if a woman works through contract employment, as is largely the case in Europe, including Poland. The average monthly earnings achieved by surgeons are up to five times what the social insurance institution offers for a pregnant woman. So, the protection is but apparent, and in such a situation, the woman’s fear becomes justified.

Sandler et al. showed that 61% of doctors responded that becoming a parent negatively affects the work of female trainees, including increasing the workload of other residents (33%) [22]. Therefore, it can be concluded that such stigmatization and negative attitude among colleagues may be the reason for resignation from the specialty or delay in gestational age. Rangel et al. showed that two of three women surveyed reported negative attitudes from both peers and superiors [3]. Moreover, a survey of medical students found that 90% of them were asked potentially discriminatory questions during interviews, including questions about marital status and having children. Fifty percent of students applying for specialty in surgery were asked about pregnancy plans, while only 14% of students applying for positions outside of surgery were asked the same question [4].

A significant difference of opinion can be seen between the group of doctors with surgical and non-surgical specialties. Non-surgeons have more restrictive views on the work of PS. More often, there were statements about stopping work in the operating theatre immediately after a confirmation of pregnancy. Moreover, PS are considered irresponsible for working in the operating theatre. There was a clear difference in opinion of doctors about the work of PS in the operating theatre. Borderline statements about work ranging from heroic to irresponsible are more common. Patients were a more unanimous group and responded that PS working in the operating room is normal. This is further proof that PS may be condemned by their partners at work.

A very interesting result of the analysis is that men, both in the group of patients and doctors, significantly more often stated that PS should stop working immediately after a confirmation of pregnancy. Moreover, male doctors were significantly less likely to be operated on by PS. We have to take into account the response bias and numerical dominance of women in the study, rather than the potential gender discrimination, which is still reported [23,24]. Having children affects, in various ways, peoples’ opinion of the immediate resignation from work by PS. Among patients, the number of children correlated with faster resignation from work, while among doctors it was the opposite. This is a surprising observation, which should be confirmed in a larger group of patients in future studies.

The study has some limitations. The survey was conducted among only Polish citizens, and their opinion was examined. However, the legal regulation applies to the entire European Union, including the protection of pregnant women. There were significantly more women than men in the study, so the opinion of men may be underestimated.

## 5. Conclusions

The study showed that female surgeons may have slight concerns about how they will be perceived by colleagues and patients. However, most of them, patients significantly more often, believed that working during pregnancy is the natural course of things. Moreover, patients significantly more often declared their willingness to be operated on by PS. Additionally, doctors more often mentioned fear of financial stability as a reason for PS continuing to work. However, the legal regulation of the European Union includes the financial protection of pregnant women.

## Figures and Tables

**Table 1 ijerph-20-02265-t001:** Characteristics of respondents. (IQR—interquartile range.)

	Doctors		Patients		
	*n* = 1074	%	*n* = 657	%	*p*
Sex					
Female	1051	97.9	611	93.0	<0.001 *
Male	23	2.1	46	7.0	
Age median (25–75% IQR)	33 (31–36)		32 (28–37)		<0.001 ^
Education					
Elementary	0	0	33	5.0	<0.001 *
Vocational	0	0	81	12.3	
Higher	1074	100	543	82.6	
Residence					
Village	102	9.5	138	21.0	<0.001 *
City <50,000 inhabitants	114	10.6	143	21.8	
City 50,000–150,000 inhabitants	144	13.4	114	17.4	
City >150,000 inhabitants	714	66.5	262	39.9	
Children					
0	137	12.8	211	32.1	<0.001 *
1	408	38.0	227	34.6	
2–3	516	48.0	209	31.8	
>3	13	1.2	10	1.5	
Relationship					<0.001 *
Single	53	4.9	68	10.4	
In relationship	1021	95.1	589	89.6	

* *p*-value was calculated using chi-square test. ^ *p*-value was calculated using Mann–Whitney test.

**Table 2 ijerph-20-02265-t002:** Opinions of doctors and patients.

		Doctors		Patients		
		*n* = 1074	%	*n* = 657	%	*p* *
When should a pregnant surgeon stop working:						
In the hospital	Immediately	110	10.2	48	7.3	0.04
	Depends	964	89.8	609	92.7	
In the operating room	Immediately	378	35.2	108	16.4	<0.001
	Depends	696	64.8	549	83.6	
In the outpatient clinic	Immediately	46	4.3	30	4.6	0.78
	Depends	1028	95.7	627	95.4	
Working at the operating table of a pregnant surgeon is:						
Normal		542	50.5	506	77.1	<0.001
Heroic		300	27.9	89	13.5	
Irresponsible		232	21.6	62	9.4	
What makes a surgeon work during pregnancy?						
Fear of financial stability		830	77.3	372	56.6	<0.001
Constant need for excitement and sensation		115	10.7	52	7.9	0.06
Constant need for self-development		548	51.0	359	54.6	0.14
Fear of losing job		358	33.3	210	32.0	0.56
Fear of losing reputation		244	22.7	103	15.7	<0.001
Fear of losing patients		122	11.4	76	11.6	0.89
Would you be operated on by a pregnant surgeon?						
No difference		961	89.5	611	93.0	0.01
No		113	10.5	46	7.0	

* *p*-value was calculated using chi-square test.

**Table 3 ijerph-20-02265-t003:** Opinions of surgeons and non-surgeons.

		Non-Surgeons		Surgeons		
		*n* = 676	%	*n* = 398	%	*p* *
When should a pregnant surgeon stop working:						
In the hospital	Immediately	69	10.2	41	10.3	0.96
	Depends	607	89.8	357	89.7	
In the operating room	Immediately	261	38.6	117	29.4	0.002
	Depends	415	61.4	281	70.6	
In the outpatient clinic	Immediately	32	4.7	14	3.5	0.34
	Depends	644	95.3	384	96.5	
Working at the operating table of a pregnant surgeon is:						
Normal		320	47.3	222	55.8	0.03
Heroic		202	29.9	98	24.6	
Irresponsible		154	22.8	78	19.6	
What makes a surgeon work during pregnancy?						
Fear of financial stability		533	78.8	297	74.6	0.11
Constant need for excitement and sensation		69	10.2	46	11.6	0.49
Constant need for self-development		343	50.7	205	51.5	0.81
Fear of losing job		235	34.8	123	30.9	0.20
Fear of losing reputation		146	21.6	98	24.6	0.25
Fear of losing patients		49	7.2	73	18.3	<0.001
Would you be operated on by a pregnant surgeon?						
No difference		596	88.2	365	91.7	0.07
No		80	11.8	33	8.3	

* *p*-value was calculated using chi-square test.

**Table 4 ijerph-20-02265-t004:** Correlation between demographic factors and responders’ answers. (* thousand inhabitants; educ. education; rel. relationship).

			When Should a Pregnant Surgeon Stop Working:									Would You Be Operated on by PS?		
			In a Hospital			In the Operating Room			In the Outpatient Clinic			No Difference	No	
			Immediately	Depends	*p*	Immediately	Depends	*p*	Immediately	Depends	*p*			*p*
Patients	Sex	Female (%)	41 (6.7)	570 (93.3)	0.03	96 (15.7)	515 (84.3)	0.07	25 (4.1)	586 (95.9)	0.03	569 (93.1)	42 (6.9)	0.64
		Male (%)	7 (15.2)	39 (84.8)		12 (26.1)	34 (73.9)		5 (10.9)	41 (89.1)		42 (91.3)	4 (8.7)	
	Educ.	Elementary (%)	1 (3.0)	32 (97.0)	0.26	4 (12.1)	29 (87.9)	0.42	1 (3.0)	32 (97.0)	0.17	29 (87.6)	4 (12.1)	0.38
		Vocational (%)	9 (11.1)	72 (88.9)		17 (21.0)	64 (79.0)		87 (16.0)	456 (84.0)		74 (91.4)	7 (8.6)	
		Higher (%)	38 (7.0)	505 (93.0)		87 (16.0)	456 (84.0)		22 (4.1)	521 (95.6)		508 (93.6)	35 (6.4)	
	Residence	Village (%)	12 (8.7)	126 (91.3)	0.92	24 (17.4)	114 (82.6)	0.41	9 (6.5)	129 (93.5)	0.43	128 (92.8)	10 (7.2)	0.21
		<50. * (%)	10 (7.0)	133 (93.0)		23 (16.1)	120 (83.9)		7 (4.9)	136 (95.1)		135 (94.4)	8 (5.6)	
		50–150. * (%)	8 (7.0)	106 (93.0)		24 (21.1)	90 (78.9)		6 (5.3)	108 (94.7)		101(88.6)	13(11.4)	
		>150. * (%)	18 (6.9)	244 (93.1)		37 (14.1)	225 (85.9)		8 (3.1)	254 (96.9)		247 (94.3)	15 (5.7)	
	Children	0 (%)	6 (2.8)	205 (97.2)	0.001	33 (15.6)	178 (84.4)	0.84	4 (1.9)	207 (98.1)	0.003	197 (93.4)	14 (6.6)	0.72
		1 (%)	19 (8.4)	208 (91.6)		41 (18.1)	186 (81.9)		8 (3.5)	219 (96.5)		208 (91.6)	19 (8.4)	
		2–3 (%)	21 (10.0)	188 (90.0)		32 (15.3)	177 (84.7)		16 (7.7)	193 (92.3)		197 (94.3)	12 (5.7)	
		>3 (%)	2 (20.0)	8 (80.0)		2 (20.0)	8 (80.0)		2 (20.0)	8 (80.0)		9 (90.0)	1 (10.0)	
	Rel	Single (%)	2 (2.9)	66 (97.1)	0.14	10 (14.7)	58 (85.3)	0.68	0 (0)	68 (100)	0.06	65 (95.5)	3 (4.4)	0.38
		In relationship (%)	46 (7.8)	543 (92.2)		98 (16.6)	491 (83.4)		30 (5.1)	559 (94.9)		546 (92.7)	43 (7.3)	
Doctors	Sex	Female (%)	105 (10.0)	946 (90.0)	0.07	368 (35.0)	683 (65.0)	0.4	43 (4.1)	1008 (95.9)	0.04	944 (89.8)	107 (10.2)	0.01
		Male (%)	5 (21.7)	18 (78.3)		10 (43.5)	13 (56.5)		3 (13.0)	20 (87.0)		17 (73.9)	6 (26.1)	
	Residence	Village (%)	9 (8.8)	93 (91.2)	0.73	37 (36.3)	65 (63.7)	0.55	6 (5.9)	96 (94.1)	0.08	92 (90.2)	10 (9.8)	0.68
		<50.* (%)	10 (8.8)	104 (91.2)		40 (35.1)	74 (64.9)		6 (5.3)	108 (94.7)		105 (92.1)	9 (7.9)	
		50–150. * (%)	18 (12.5)	126 (87.5)		58 (40.3)	86 (59.7)		11 (7.6)	133 (92.4)		126 (87.5)	18 (12.5)	
		>150. * (%)	73 (10.2)	641 (89.8)		243 (34)	471 (66)		23 (3.2)	691 (96.8)		638 (89.4)	76 (10.6)	
	Children	0 (%)	21 (15.3)	116 (84.7)	0.04	44 (32.1)	93 (67.9)	0.3	11 (8.0)	126 (92.0)	0.09	121 (88.3)	16 (11.7)	0.95
		1 (%)	47 (11.5)	361 (88.5)		156 (38.2)	252 (61.8)		18 (4.4)	390 (95.6)		365 (89.5)	43 (10.5)	
		2–3 (%)	42 (8.1)	474 (91.9)		172 (33.3)	344 (66.7)		17 (3.3)	499 (96.7)		463 (89.7)	53 (10.3)	
		>3 (%)	0 (0)	13 (100)		6 (46.2)	7 (53.8)		0 (0)	13 (100)		12 (92.3)	1 (7.7)	
	Rel.	Single (%)	7 (13.2)	46 (86.8)	0.47	20 (37.7)	33 (62.3)	0.69	2 (3.8)	51 (96.2)	0.85	50 (94.3)	3 (5.7)	0.24
		In relationship (%)	103 (10.1)	918 (89.9)		358 (35.1)	663 (64.9)		44 (4.3)	977 (95.7)		911 (89.2)	110 (10.8)	

## Data Availability

Not applicable.

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
