# Peer review of "To Be a Pregnant Surgeon—Is There Anything to Be Afraid of?"

_ijerph, 2023, doi:10.3390/ijerph20032265_

Round 1
Reviewer 1 Report
The article is very interesting. It presents interesting results of surveys on the problem of pregnant surgeons from the point of view of patients and doctors.
The caveat is the very limited "Introduction" section. There is a lack of literature review on the potential risk of complications during pregnancy, taking into account the toxicity of anesthetic gases to operating theatre medical personnel, the constrained positioning of operators (often for many hours), stress factors, and the need to use force in some types of operations.
Many sentences included in the Discussion should be moved and expanded in the Introduction. The topic of occupational safety of pregnant surgeons was also addressed by other authors. I did not find citations of the following articles in the References:
Pregnant surgeon - assessment of potential harm to the woman and her unborn child. Szczesna A, Grzelak K, Bieniasz M, Kacperczyk-Bartnik J, Dobrowolska-Redo A, Bartnik P, Zareba-Szczudlik J, Romejko-Wolniewicz E.Ginekol Pol. 2019;90(8):470-474. doi: 10.5603/GP.2019.0081.PMID: 31482551
Safety of radioactive sentinel node biopsy for breast cancer and the pregnant surgeon - A review. Saha S, Jacklin R, Siddika A, Clayton G, Dua S, Smith S.Int J Surg. 2016 Dec;36(Pt A):298-304. doi: 10.1016/j.ijsu.2016.11.019. Epub 2016 Nov 10.PMID: 27840311
Occupational hazards to the pregnant orthopaedic surgeon. Keene RR, Hillard-Sembell DC, Robinson BS, Novicoff WM, Saleh KJ.J Bone Joint Surg Am. 2011 Dec 7;93(23):e1411-5. doi: 10.2106/JBJS.K.00061.PMID: 22159863
Radiation exposure to a pregnant urological surgeon – what is safe? Birnie AM, Keoghane SR.BJU Int. 2015 May;115(5):683-5. doi: 10.1111/bju.12923. Epub 2015 Jan 21.PMID: 25195709
In the Discussion, the authors should refer more to the results obtained in the questionnaire. It is worth analyzing the reasons for the different views of doctors and patients on the working conditions of pregnant surgeons. It will certainly be related to the demographic data of the studied groups.
It would be worth emphasizing in the Conclusions that the main reason for the work of surgeons during pregnancy is for physicians (results in tables 2 and 3) the fear of financial stability (830 out of 1074 doctors, including 74.6% of 398 surgeons, gave this answer). In the eyes of patients, this reason was important, but no less respondents also indicated the answer “Constant need for self-development”.
Author Response
Thank you very much for taking the time for the review and your valuable comments. Our answers are attached in the word file.

Reviewer 2 Report
Introduction
The introduction makes several broad statements that require supporting evidence or should be revised to be less definitive. For example - The authors imply that the introduction of minimally invasive surgery is what led the shift from male predominance to a growing proportion of women surgeons – is this true? Similarly, the statement that “Women who decide to become a surgeon are afraid of motherhood” is too strong and general.
I have not seen evidence that “is has been proven that some instruments and operating environment have a negative impact on pregnancy” despite writing a scoping review on this topic – the authors should clarify what instruments or environmental factors, and what types of adverse outcomes (e.g., preterm birth? Intrauterine growth restriction?) are associated. This should be cited, otherwise this statement will perpetuate fear or rumor about surgery and pregnancy.
The authors state that a secondary objective of their study was to review current maternity policy but the Methods and Results do not contain any information about this objective, including how this was done or what the results of this review were.
Methods
The authors should provide more details about the survey development, distribution, and analysis. They could use the EQUATOR-CHERRIES checklist to see what details of survey methods should be included. For example, what social media accounts were used to advertise the survey? Who is the audience of these accounts? Were these institutional social media or personal social media accounts? This will help the reader understand potential response bias.
The authors state that beginning the survey was considered to be consent – was this approved by an ethics review board?
Results
The authors should not combine the responses for participants who thought being operated on by a pregnant surgeon is heroic with those who think it is irresponsible – these are opposite beliefs with presumably different underlying constructs. For example, heroic implies that the physician is overcoming something difficult and irresponsible implies that the surgeon is harming someone needlessly.
The authors should make a better case for why the opinions of not pregnant, non-surgeons on the financial stability or decision to keep working of pregnant surgeons are important – why does it matter that patients think that financial stability of an operating surgeon is a motivator? Why does a patient decide when a pregnant surgeon stop operating? How does this result inform the study question? In contrast, the finding that patients don’t care whether their surgeon is pregnant is important – because it addresses the argument that it is not good for patients.
To me, the differences in section 3.2 are the most interesting – in particular, the differences between men and women physicians in opinions about working, pregnant surgeons. This suggests an element of maternal discrimination in how men view working, pregnant colleagues.
Discussion
The study results do not support the statements made in the Discussion – in particular, that “pregnancy among surgeons should be a reason for resignation from work in the operating theatre”. The authors imply that the opinions of non-pregnant patients and non-pregnant physicians should determine the actions of working, pregnant surgeons, which is not only false but completely harmful.
Author Response

(The authors gave the same response as above.)

Reviewer 3 Report
This study describes perspectives about pregnant surgeons both from doctors and patients using the same questionnaire. As this type of paper is unique, it is better to be published. However, there are some concerns which are required to resolve.
1. Authors conclude that female surgeons may have legitimate concerns about how they will be perceived by colleagues and patients. However, patients are even favorable to PS than female surgeons (Working at the operating table of a pregnant surgeon is normal: female surgeon 55.8% vs. patient 77.1%). Conclusion should reflect the results you obtained.
2. In the methods section, the descriptions of respondents and the way how survey was distributed are too rough. Study population should be precisely described. What kind of SNS plat form were used and numbers of registered members (numbers of doctors and patients supposed to be sent ) should be written. Institutional review board statement must also be clearly written.
3. Although occupational hazards are mentioned in the introduction and discussion, there are no results indicating occupational hazards.
4. Table 4 is too complicate to read.
Author Response

(The authors gave the same response as above.)

Round 2
Reviewer 3 Report
I noticed some changes in the manuscript.
However, there are still some portions which need to be cleared.
1. Although the number of respondents were many, there might be significant bias because we cannot know the response rate or what kind of patients responded to this survey. You should mention about this matter.
2. It is still unclear why doctors including surgeons thought PS should stop operation immediately.
3. Table 4 is still really busy. Also, even if there was a difference in each item, if you assess comprehensively, it is doubtful the difference was haphazard or not.
4. Although the occupational hazard was mentioned in introduction section, it was not discussed according to the results.